# Effect of Trough Incidence Angle on the Aerodynamic Characteristics of a Biomimetic Leading-Edge Protuberanced (LEP) Wing at Various Turbulence Intensities

**DOI:** 10.3390/biomimetics9060354

**Published:** 2024-06-12

**Authors:** Shanmugam Arunvinthan, Ponnusamy Gouri, Saravanan Divysha, RK Devadharshini, Rajan Nithya Sree

**Affiliations:** School of Mechanical Engineering, SASTRA University, Thanjavur 613401, Tamil Nadu, India; gourip0303@gmail.com (P.G.); divyshas@gmail.com (S.D.); dharshinid572@gmail.com (R.D.); 2004nithyar@gmail.com (R.N.S.)

**Keywords:** leading-edge protuberances, trough incidence angle, aerodynamic force coefficients, surface-pressure characteristics, pressure measurement, wind tunnel testing, pressure-integration technique, biomimetics and turbulence intensities

## Abstract

A series of wind tunnel tests were performed to investigate the effect of turbulent inflows on the aerodynamic characteristics of variously modified trough incident leading-edge-protuberanced (LEP) wing configurations at various turbulence intensities. A self-developed passive grid made of parallel arrays of round bars was placed at different locations of the wind tunnel to generate desired turbulence intensity. The aerodynamic forces acting over the trough incidence LEP wing configuration where obtained from surface pressure measurements made over the wing at different turbulence intensities using an MPS4264 Scanivalve simultaneous pressure scanner corresponding to a sampling frequency of 700 Hz. All the test models were tested at a wide range of angles of attack ranging between $0^{{}^{\circ}}\leq \alpha \leq 90^{{}^{\circ}}$ at turbulence intensities varying between 5.90% ≤ TI ≤ 10.54%. Results revealed that the time-averaged mean coefficient of lift (C_L_) increased with the increase in the turbulence intensity associated with smooth stall characteristics rendering the modified LEP test models advantageous. Furthermore, based on the surface pressure coefficients, the underlying dynamics behind the stall delay tendency were discussed. Additionally, attempts were made to statistically quantify the aerodynamic forces using standard deviation at both the pre-stall and the post-stall angles.

## 1. Introduction

Striving towards sustainable and renewable energy sources has become crucial in addressing the global challenges of climate change and depleting fossil fuel reserves. Among the diverse array of renewable energy technologies, wind power stands as a prominent and environmentally benign option. Wind turbines have rapidly evolved in recent years, with advancements in materials, design, and efficiency. However, as wind energy technologies continue to mature, so too does the need for further innovation to increase energy capture and turbine performance while minimizing environmental impacts. Most of the repowering strategies in the wind turbines focus their attention on repowering/replacement of older generation turbines with newer generation more efficient turbines and in some cases hub-height increment. It should however be noted that repowering efforts should also include the wind turbine blades by altering/modifying the older generation blades to the modern-day efficient blades by incorporating features that can augment the power performance of the wind turbines. Nature has long provided inspiration for human innovation, and the field of biomimetics, or biomimicry, seeks to harness the wisdom of evolution for human applications. In this context, the natural world’s aerodynamic marvels offer a promising source of inspiration for optimizing wind energy technology. One such area of particular interest is the concept of leading-edge protuberances (LEPs) inspired from the intricate wing structures of humpback whales’ pectoral flippers. The leading edge of the humpback whale is a testament to evolution’s mastery of aerodynamics.

Nature has finely tuned avian wing structures, such as feathers and leading-edge serrations, to reduce drag, improve lift, and enhance overall flight performance. This intricate adaptation has enabled whales to navigate diverse and often turbulent atmospheric conditions efficiently. Drawing inspiration from nature is not new for the aerodynamic and wind engineers. Such unique leading-edge protuberances found on the flippers of the humpback whale was first noticed by Frank. E. Fish, a marine biologist. Intrigued by its uniqueness, Fish et al. [1] investigated the morphology of the flippers from a beached humpback whale and reported that the leading edge of the humpback whale flipper is characterized by a series of protuberances along its surface. It was also identified that irrespective of the spanwise position, the cross-section of the flipper remains constant which resembles the man-made NACA 63(4)-021 airfoil section. Watts and Fish [2,3] performed numerical investigation over the NACA 63(4)-021 airfoil section with and without leading-edge protuberances. The computational results were quite promising with a lift increment of 4.8% and a drag reduction of 10.9% thus leading to an overall aerodynamic efficiency increment of 17.6%. Subsequently, Watts and Fish patented this technology and started a company called Whale Power Corporation, Toronta, Canada. Frank E. Fish continued his research and subsequently published several research articles [4,5,6,7,8]. Miklosovic et al. [9] conducted one of the earliest wind tunnel experiments on a scaled model of an idealized humpback whale flipper with and without tubercles. Experimental results revealed that the modified flipper model with tubercles not only augmented the aerodynamic lift performance characteristics but also offered a stall delay of 40% without any additional drag penalty. Downer and Dockrill [10] of the Wind Energy Institute of Canada tested the Whale Power tubercled wind turbine blade and revealed that the modified model with tubercles possessed 25% more airflow than the unmodified equivalent and produced 20% more energy. It is speculated that these unique leading-edge protuberances on the humpback whale act like a flow control mechanism thus offering exceptional maneuvering capabilities during its co-operative feeding method called bubble net feeding. Since the researchers identified that the amplitude and the wavelength of the leading-edge protuberances were unique and were constantly changing from the root to the tip of the flippers, the researchers focused their attention on identifying the optimum amplitude and wavelength.

Johari et al. [11] experimentally evaluated the effect of various leading-edge geometries on the aerodynamic performance characteristics of the finite span wings on different planforms like rounded tip, swept-leading edge planform, and an idealized flipper planform at Re = 1.83 × 10^5^. Johari et al.’s [11] selection of amplitude and wavelength placed their work in a strategically important position in addressing the amplitude and wavelength variation. Based on the morphology found on the leading-edge of the humpback whale flippers, Johari et al., reported that the amplitude of the leading-edge protuberances varied from 2.5% to 12% of the mean chord length and spanwise wavelengths were 25 to 50% of the mean chord length. Experimental results revealed that the wavelength of the protuberances and the leading-edge radius played a minor role while the amplitude of the protuberances played a significant role in the aerodynamic forces and moments. The lift-to-drag ratio for the post-stall regimes exhibited 50% improvement over the baseline case. The effect of waviness was further investigated by Yoon et al. [12] for five different waviness ratios like 0.2, 0.4, 0.6, 0.8, and 1.0 keeping the amplitude and the wavelength constant. Test foils featuring a waviness ratio of 1.0 exhibited better performance and outperformed the baseline clean wing configuration indicating that the leading-edge protuberance needed to be present over the entire span. Kim et al. [13] extended the study conducted by Yoon et al. [12] by investigating five different wavelengths for fixed amplitude. Experimental results indicated the aerodynamic force coefficients like ($C_{L}$) and ($C_{D}$) were almost negligible with the variation in the wavelength. This result held in good agreement with the Johari et al.’s [11] findings. Arunvinthan et al.’s [14] recent research confirmed these findings and reported that the effect of the amplitude had a significant effect when compared against the wavelength. Their experimental evaluation indicated that LEP with amplitude 0.12c exhibited better aerodynamic performance than 0.025%C amplitude featuring a common wavelength of 0.5%C. Aiming at identifying the significance of LEP in the finite and infinite wing configurations, Kouh et al. [15] investigated the effect of amplitude and wavelength on finite and infinite wing planform geometries. Post-stall angle results indicated that leading-edge protuberanced wings outperformed the baseline wing in both cases, thereby clearly indicating that the incorporation of leading-edge protuberances was beneficial for finite as well as infinite wing configuration. Pedro and Kobayashi et al. [16] previously suggested that the higher aerodynamic performance of the modified LEP wing was due to the compartmentalization of the flow. Fernandes et al. [17] conducted extensive experimental and numerical research to gain an insight in to the underlying flow physics. It was found that at $\alpha =12^{o},$ both the baseline clean wing configuration and the modified leading-edge protuberanced wing showed boundary layer growth, while the modified leading-edge protuberanced wing showed asymmetrical flow patterns i.e., the flow remained attached over the peak region of the modified wing while flow separation was observed at trough. Vân Nierop et al. [18] reported that the bump on the leading edge of the model humpback whale flippers caused them to ‘‘stall’’ i.e., lose lift dramatically, more gradually, and at a higher angle of attack. Their aerodynamic model suggested that the presence of the leading-edge tubercles changed the pressure distribution across the airfoil (i.e., non-uniform separation characteristics). It was reported that the downwash was larger at the peak section relative to the troughs due to the non-uniform separation characteristics, leading to a decrease in the effective angle of attack further delaying stall for the peak section. So, it became clear from the Nierop’s findings that the LEP wings outperformed the conventional wings at post-stall angles of attack. The hydrogen bubble flow visualization study performed by Hansen et al. [19] revealed that the leading-edge protuberances generated vortices which a change in the direction of rotation with respect to the change in the leading-edge geometry which tended to cancel out each other resulting in a shorter wake creating no additional drag penalty.

Custodio et al. [20] identified that the performance parameters were highly dependent on the tubercle amplitude based on previous studies, however, the performance parameters variation in terms of planform geometries remained unclear. Custodio et al. [20] tested four different planform geometries like finite span, infinite span, swept, and idealized flipper model and reported that the span efficiency factor decreased for all the leading-edge-modified models when compared to the baseline models and the swept wing leading-edge-modified airfoil appeared to be the most affected. The drag of this planform was appreciably greater than the baseline model throughout the angle of attack range and the lift was reduced in the pre-stall regime. Even though the leading-edge modification using LEP provided better results in the post-stall angle, the reduction in lift in the pre-stall regime is an issue which needs to be addressed immediately. Custodio et al. [20] concluded that the effectiveness of the LEP was dependent on the planform geometry and the maximum lift coefficient of the rectangular planform models was 10% greater than the baseline model. Skillen et al. [21] focused their attention on the reason behind the asymmetrical flow pattern on the rectangular LEP wing and elucidated the observed mechanism by which the LEP renders aerodynamic benefit as follows: “Oncoming free-stream flow is bifurcated by the leading-edge protuberances in such a way that the bulk of the flow goes behind the trough region when compared against the peak region. This leads to a strong acceleration behind the trough consequently forms an enhanced suction peak driving the development of a secondary flow owing to the spanwise pressure gradient”. This enhanced suction draws the low-inertial boundary layer fluid from the peak which is then replaced by the high momentum fluid drawn from above which energizes the flow at the peak, delaying separation behind the peak. On the other hand, the effect of the suction peak-drawn low-inertial boundary layer fluid in addition to the reduced chord length leads to a large adverse pressure gradient result in flow separation at short distance downstream of the leading-edge at the trough. This alternative flow pattern could plausibly explain the reason behind the compartmentalization of the flow. Hansen et al. [22] experimentally evaluated the leading-edge-protuberanced models under the water tunnel facility and confirmed the formation of streamwise vortices behind the leading-edge protuberances using PIV results. It was also reported that the streamwise vortices generated over the airfoil due to the presence of the LEP improved boundary layer momentum exchange. Aftab et al. [23] summarized the fact that the LEP wing showed increased lift by delaying and restricting spanwise separation. Furthermore, his research results suggested that leading edge protuberances produced vortices by deflecting the fluid that was flowing through them into the troughs as confirmed by Skillen et al. [21]. The fluid in the peak region interacted with these vortices, energizing the flow. The streamline contours, the normalized vorticity contours, and even the oil flow visualization showed the mixing of vortices along both the chordwise and the spanwise directions. The early laminar separation on the peak and its merger in to the turbulent flow in the trough created a complicated flow thus modifying the separation as reported by several researchers [11,20,21,24].

Although several explanations for the performance enhancement observed in the LEP like (a) boundary layer momentum exchange induced by chordwise and streamwise vortices, (b) flow compartmentalization, and (c) non-uniform separation characteristics due to alteration of surface pressure distribution etc., are reported. It is clear that the surface pressure characteristics of the LEP section varies primarily based on the amplitude and the wavelength which had a negligible effect as outlined by previous researchers. It should, however, be noted that they are not the only two parameters which influence the performance characteristics of the LEP as they are also a function of angle of attack (i.e.,) incidence of the peak and the trough region of the LEP relative to the freestream wind. Therefore, this paper focused on investigating the influence of the incidence angle on the performance characteristics of LEP. Previous studies outlined that a pair of streamwise vortices exist as counter rotating pairs in the trough between the protuberances. Custodio et al. [20] suggested that their formation was a result of spanwise flow associated with the protuberances as shown by Skillen et al. [21]. It is believed that the vortices lead to the additional momentum exchange in the boundary layer thereby providing the aerodynamic benefit in terms of delayed stall characteristics. This clearly shows that the aerodynamic benefit offered by the LEP is most likely due to the spanwise flow. Moreover, at the same time, it is of interest to note that this spanwise flow is the reason behind the lower coefficient of lift associated with the LEP models. That is because this spanwise flow over the LEP directly influences the net pressure gradient existing between the pressure and the suction side of the airfoil which is expressed in terms of lift force. At zero incidence angles, where the peak and the trough of the LEP is encountering the freestream flow, the peak experiences the early laminar separation and its merger in to the turbulent flow in the trough as the low-inertial boundary layer molecules were drawn from the peak to the trough caused by the enhanced suction created by the spanwise flow. It becomes clear that if the peak or trough incidence angle varies, the spanwise flow will not be the same, thereby effectively altering the surface pressure characteristics existing over the LEP model.

The effect of incidence angle on the LEP is a new phenomenon which is not addressed by any researchers around the world and is very crucial since it can alter the flow characteristics over the LEP model. To the extent of the authors’s knowledge, the effect of the incidence angle on the LEP wing remains unclear and untouched. Therefore, the present paper aimed at addressing this issue in terms of trough incidence angle $\left(\theta_{T}\right)$. Since the spanwise flow induced between the peak and the trough section of the LEP is a complicated flow, the influence of the turbulence intensity on such complicated flow also needed to be addressed. Previous literature confirmed that the turbulence intensity significantly affects the aerodynamic performance of leading-edge protuberance airfoils. Stack et al. [25] initiated the study on the effect of turbulence intensity on the aerodynamic forces acting over various airfoil models in the presence and absence of a turbulence grid in a wind tunnel and found that the increase in turbulence intensity results in an increase in (C_L,max_) and effectively delays the stall. On the other hand, Li et al. [26] suggested that higher turbulence levels can hinder flow separation on the surface of the airfoil. Aiming at identifying the influence of turbulence intensity on the aerodynamic characteristics of the variously modified leading-edge protuberanced wing configuration, Arunvinthan et al. [14] experimentally evaluated the LEP wings at various TI. Results revealed that the complicated flow pattern induced by the spanwise flow persisted in the presence of turbulence intensity and continued to offer the stall delay benefit at zero incidence angles, however, the influence of the aerodynamic characteristics of the trough incident LEP is still lacking. In this paper, the experimental results of trough incidence angle $\left(\theta_{T}\right)$ on the aerodynamic characteristics of the variously modified leading-edge-protuberanced wings at different turbulence intensities obtained using a Scanivalve MPS4264 simultaneous pressure scanner are reported. The result showed the variation of aerodynamic force coefficients like the coefficient of lift (C_L_) and coefficient of drag (C_D_) of the variously modified leading-edge-protuberanced models at different turbulence intensities along with their corresponding surface pressure distribution. Understanding the aerodynamic characteristics of the trough-incident leading-edge-protuberanced wing along with the influence of TI will be highly beneficial and could be advantageous for aerodynamic designers and wind engineers to bring such novel design into practical existence to enhance maximum wind capture at wider operating ranges yielding aerodynamic robustness. Future work should address the peak incidence angle $\left(\theta_{P}\right)$ and the potential role of incidence angle if these are found to be present.

## 2. Experimental Methodology

### 2.1. Synthesis of Experimental Setup

The aerodynamic performance characteristics of trough incident leading-edge protuberanced test models were experimentally evaluated in a low-speed subsonic wind tunnel facility located at SASTRA Deemed University. The rectangular test-section of the wind tunnel was 300 mm × 300 mm in the cross-section with the length of 1500 mm. The tunnel was operated by a fan powered by a 10 HP motor that could reach a maximum wind velocity of round 60 m/s. A schematic representation of the experimental apparatus involved in this paper is shown in Figure 1. All the test models considered in this study were based on NACA 63(4)-021 airfoil since it closely resembles the flippers of humpback whales. Experiments were performed at four different Reynolds numbers ranging from $0.96\times 10^{5}$ to $3.5\times 10^{5}$ corresponding to mean free-stream velocities from 13.7 m/s to 50 m/s. It should, however, be noted that the authors had some experimental constraints as well. In this experimental facility, even if the wind tunnel tests were performed at the maximum speed of 60 m/s, the maximum possible Reynolds number was 3.97 × 10^5^. To increase the Reynolds number range, another possibility was to consider a test model of higher chord length. Some of the rough calculations are shown here for the detailed explanation:

Test model of chord 150 mm–(@ 60 m/s)—Max. Re was 5.95 × 10^5^

Test model of chord 200 mm–(@ 60 m/s)—Max. Re was 7.94 × 10^5^

Test model of chord 250 mm–(@ 60 m/s)—Max. Re was 9.92 × 10^5^

Test model of chord 275 mm–(@ 60 m/s)—Max. Re was 1.09 × 10^6^

Based on the values, it became clear that to reach the Reynolds number regime in the order of 10^6^, the chord should be at least 275 mm. If the test model of chord 275 mm was utilized, when inclined at higher angles of attack, it would block the flow in the wind tunnel, thus creating a blockage effect, practically making the results unusable. Henceforth, the test model chord was restricted to 100 mm thus leading to the Reynolds number in the order of 10^5^. Similarly, the same model was used throughout the study and different geometries were not used to change the Reynolds number due to financial constraints. Based on the previous literature, it became clear that the modified leading-edge protuberanced model tended to offer stall delay characteristics for an extended range of angles of attack. Henceforth, in the present study, experimental evaluations were performed over all the test models for a wide range of angles of attack ranging from 0° to 90° in an increment of 5°.

### 2.2. Leading-Edge Protuberanced Wing Model Configurations

A baseline leading-edge protuberanced model featuring a 0.12c amplitude(A) and a 0.25c wavelength(λ) inspired from the previous investigation [14] was chosen to be the test model. Aiming at identifying the influence of the trough incidence angle, two different test models featuring trough incidence angles of 4° and 8° were 3D Printed. A schematic representation along with the original 3D-printed test model is shown in Figure 2. It should, however, be noted that the current study aimed only to investigate the effect of trough incidence angle on the aerodynamic characteristics of the leading-edge protuberanced (LEP) wing alone and not the optimal effect of leading-edge protuberances amplitude(A) and wavelength(λ). A NACA 63(4)-021 airfoil was considered for this study as it closely resembles the flippers of the humpback whales as outlined by Fish et al. [1].

All the LEP test models considered in this study were full-span rectangular models featuring a mean aerodynamic chord (c) of length 100 mm and span (s) 300 mm, fabricated via 3D printing using polylactic acid or polylactide material at a resolution of 100 µm. Followed by 3D printing, the models were subsequently post-processed using a sequence of abrasive sheets to obtain an ultra-marble finish. As mentioned earlier, the geometrical parameters of the LEP like amplitude (A) and wavelength (λ) were kept constant as 0.12c and 0.25c based on the previous literature as outlined in Arunvinthan et al. [14]. Aiming at identifying the effect of trough incidence angles, two trough incidence angles, namely 4° and 8°, were considered as tabulated in Table 1. In order to measure the surface pressure, a total of 50 pressure tapings were made over the suction and the pressure side of the airfoil along the peak and trough sections of the baseline LEP as well as trough incident LEP models. Generally, in a conventional airfoil, the pressure tapings are distributed over the suction and the pressure side of the airfoil, however, with the LEP model being a non-constant-chord model, the location of the surface pressure taps had a special consideration. Because of the variation of chord over the span, an LEP model requires surface pressure measurement over the peak and the trough locations. This places a severe restriction on the total number of tapings located on the model. In this present study, LEP models featuring 50 pressure taps were utilized. All the 50 pressure taps were equally-distributed among the suction and the pressure side of the models as shown in Figure 3. Each pressure taping was approximately 1 mm in diameter and the distance between each surface pressure tap was maintained at ≤0.1c for better spatial representation. The peak and trough incidence angles further complicated the pressure taping near the vicinity of the trailing edge by creating a vertical zig-zag pattern.

### 2.3. Simultaneous Time-Series Pressure Measurement

Aiming at measuring the surface pressure distribution over the baseline leading-edge protuberance models and the modified trough incident LEP models, a 64-channel MPS4264 Scanivalve simultaneous pressure scanner was utilized. As mentioned in Section 2.2, a total of 50 surface pressure taps were equally distributed over the pressure and the suction side of the test models along the peak and the trough section for both the baseline and the modified models. Those pressure tapings from the models were then pneumatically connected to the pressure scanner through high-quality polyurethane tubes. MPS4264 is a miniaturized piezo-resistive pressure sensor capable of measuring 10” of H_2_O at a sampling frequency of 850 Hz in binary mode and up to 2500 Hz in fast mode, weighing only 217 g. It also houses four digital temperature chips and four RTDs for accurate temperature compensation. In this present study, the time-series surface pressure data over both the baseline and the modified LEP models featuring peak and trough incidence angles were collected at a sampling frequency of 700 Hz corresponding to 10,000 data samples. The aerodynamic forces like lift and drag acting over the test models were then obtained by integrating the surface pressure data over the entire surface based on the pressure integration technique. The coefficient of lift and drag was then derived from the lift and the drag forces as outlined by Arunvinthan et al. [14]. Two-dimensional infinite test models were considered to avoid any three-dimensional flow and hence, the flow changes experienced by the test models were purely due to the variation of the trough incidence angles alone. It is of interest to note that the uncertainties in the case of pressure measurement is addressed in Section 2.4.
(1)$$F_{L}=\sum \left(P_{i}-P_{0}\right)\times \sin\left(\theta_{i}+\alpha \right)$$
(2)FD=∑Pi−P0×cos ⁡θi+α
(3)$$C_{L}=F_{L}/0.5\rho CV^{2}$$
(4)$$C_{D}=F_{D}/0.5\rho CV^{2}$$

A self-developed passive grid made up of parallel arrays of round bars was utilized in this study to create the desired turbulence intensity. In this study, the grids were placed at different locations upstream of the model and their corresponding turbulence intensities were measured using a seven-hole probe of M/S, (Aeroprobe corporations, Christiansburg, VA, USA). The turbulence intensity considered in this study was a one component TI in the x-direction and the flow was isotropic and homogeneous. Repeated measurements were made at the model location to ensure the stability of the measurements and the maximum error was 1.17%. For a detailed discussion on the experimental setup related to turbulence intensity, please refer to our earlier study Arunvinthan et al. [14] which explored the foundation framework.

### 2.4. Uncertainty Analysis

The uncertainty associated with the experiments and their corresponding corrections are listed in this section.

#### 2.4.1. Buoyancy

The variation of static pressure along the test section was induced in the closed throat wind tunnel because of the thickening of the boundary layer as it progressed towards the exit cone resulting in effective diminution of the jet area when no model was present; it was non-zero in many wind tunnels and this can produce a drag force hence a suitable correction needed to be implemented over the readings. However, it should be noted that it should not be confused with the static pressure variation along the wind tunnel that was induced by the presence of the test model. Glauert’s method of horizontal buoyancy correction was implemented in the present study. Glauert reported that the total drag increment due to the horizontal buoyancy for a two-dimensional infinite model can be estimated as follows
(5)$$D_{B}=\frac{1}{2}\pi \lambda_{2}t^{2}P\prime$$
where *λ*_2_ is the body shape factor, *t* is the body thickness and *P*′ is the slope of the longitudinal static pressure gradient curve. The body shape factor for the NACA 63(4)-021 airfoil utilized in this study was obtained as 1.4 as outlined in Barlow et al. [27]. The drag due to Buoyancy was estimated as 0.009693 for this paper.

#### 2.4.2. Solid Blockage

The ratio of the frontal area of an article to the stream cross-sectional area is effectively zero in most actual operations but in wind tunnel tests, this ratio reflects the relative size of the test article and the test section as the wind tunnel walls confines the flow around the model. Based on the framework of the previous research and outlined by Barlow et al., the solid blockage is usually chosen in the range of 0.01 to 0.10 with 0.05 being typical. These values typically indicate the effective change in the oncoming flow speed or dynamic pressure due to the blockage created by the model. The presence of tunnel walls reduces the area through which the air must flow as compared to the free-air conditions. This reduction in the area increases the velocity of the air as it flows in the vicinity of the model. Such solid blockage is a function of model thickness, thickness distribution, and the model size. The present paper utilized the simpler form of solid blockage correction for two dimensional tunnels proposed by Thom. Thom’s solid blockage correction is
(6)$$\varepsilon_{sb}=\frac{K_{1}\;(Model\;Volume)}{C^{\frac{3}{2}}}$$
where $K_{1}$ equals 0.74 for wing spanning the tunnel breadth and 0.52 for one spanning the tunnel height; C stands for test-section area. The model volume can be approximated as follows:(7)Model Volume=0.7×model thickness×model chord×model span

The model volume was estimated to be 0.441 and the solid blockage was estimated to be 0.01208 for this paper.

#### 2.4.3. Wake Blockage

This effect is a result of the finite size of a body wake, but it is more complicated because the size of the wake itself is a function of body shape and the ratio of the wake area to the tunnel area. The two-dimensional wake blockage correction provided by Allen and Vincentti was utilized in this paper. Maskell examined the effect of flow outside the wake and explained how its higher speed results in a reduced pressure over the rearward portion of the model. The wake blockage correction factor for the two-dimensional case is shown below.
(8)$$\varepsilon_{wb}=\tau \left(c_{du}\right)$$
where $c_{du}$ is the uncorrected drag coefficient and $\tau =\frac{c/h}{4}$. Studies indicate that Maskell’s paper also yields the same relation as outlined by Barlow et al. [27]. Allen and Vincentti also provided their correction factor involving the body shape factor of the airfoil and hence, it was deemed to be more accurate for this study as it involves the airfoil section.
(9)$$∆C_{d,wb}=\wedge \sigma$$
where, ∧ denotes the body shape factor which was 0.36 for the NACA 63(4)-021 airfoil as outlined in Barlow et al. [27]. However, σ can be estimated from $\sigma =\left(\frac{\pi^{2}}{48}\right){\left(\frac{c}{h}\right)}^{2}$; for the present study it was estimated to be 0.0616225 and the overall wake blockage correction was estimated to be 0.0221841.

##### 2.4.4. Instrument and Data Error

Instrument and data error studies indicate that most of the uncertainties involved in any experiment are largely caused by the instruments. Since the instrument involved in the present experiment for pressure measurement was precise with a full-scale error of ±0.06%, any correction implemented for the uncorrected value only affected the third decimal. Likewise, dispersion of data was one such uncertainty but measurements were repeated for several test cases and subsequently it was found that the time-averaged values did not encounter any significant uncertainty. One of the recent studies by Li et al. [26] confirmed this statement by making a claim that uncertainties due to the dispersion of data could be easily overcome by a large number of samples.

## 3. Results and Discussion

The Effect of TI on the aerodynamic characteristics of 4° and 8° trough incident on the LEP Model can be seen.

(1)*Re* = *0.96* × *10*^5^

Figure 4 illustrates the time-averaged coefficient of lift (C_L_) against the angle of attack (α) for the 4° trough incident leading-edge protuberanced (LEP) model at various turbulent intensities ranging from 5.90% to 10.54% at Re = 0.96 × 10^5.^ It can be clearly seen from the figure that the time-averaged coefficient of lift trendline for the 4° trough incident LEP model subjected to TI = 5.90% was slightly higher than the lift coefficient of the same LEP model subjected to a turbulence intensity of 6.85% especially when α > 15°. For instance, at α = 25°, the 4° trough incident LEP model subjected to a TI of 5.90% had a lift coefficient of 0.30 whereas the same 4° trough incident LEP model subjected to a TI of 6.85% produced a lift coefficient of 0.27 which was 3% lower than its counterpart at TI = 6.85%. But with the further increase in the turbulence intensity, the coefficient of lift started exhibiting higher values than the model subjected to lower magnitudes of turbulence. For instance, it can be seen from the figure that the time-averaged coefficient of lift of the 4° trough incident LEP model subjected to 7.55% was higher than the TI = 6.85%. As shown in the figure, the maximum lift coefficient of the 4° trough incident LEP model occurred at α = 50°, especially at higher angles beyond α = 35° where the lift coefficient values were more pronounced. An increasing trend in the maximum lift coefficient with the increase in the turbulence intensity could be observed up to TI = 7.55% beyond which the adverse effects started appearing. It is believed that the complicated spanwise flow pattern existing between the peak and trough region gets influenced by the introduction of the trough incident angle which further complicates the flow pattern over the airfoil, thereby resulting in the detrimental characteristics. The maximum lift coefficient of the 4° trough incident LEP model was 0.26, 0.27, and 0.29 for TI = 5.90, 6.85, and 7.55%, respectively, whereas the same 4° trough incident LEP model subjected to 8.49% performed apparently poorer than the TI = 7.55% case. Similarly, the coefficient of lift for TI = 10.5% performed poorer than the TI = 7.55% case. This clearly showed that at Re = 0.96 × 10^5^, the lift coefficient of the 4° trough incident LEP model increased with the increase in the TI from 5.90% to 7.55%. Aiming at identifying the influence of the turbulence intensity on the stall characteristics of the 4° trough incident LEP model, the lift curve characteristics were further explored. It was evident from the figure that for the 4° trough incident LEP model subjected to TI = 5.90%, the coefficient of lift linearly increased with the increase in the angle of attack up to α = 10°. Beyond this, it continued to increase lift to α = 25° and then underwent the stall phenomenon accompanied with the loss of lift. As it is known from the fact that the LEP model did not undergo complete stall, it could be seen from the figure that the 4° trough incident LEP model started producing lift at α = 30° and continued to produce lift up to α = 55° which was significantly higher than the conventional stall angles. However, with the increase in the turbulence intensity as the flow characteristics underwent transition, the stall characteristics of the 4° trough incident LEP model also underwent changes. It can be seen from the figure that for the TI = 6.85% and 7.55% cases, the time-averaged coefficient of lift continued to increase with the increase in the angle of attack up to α = 25° and then underwent a slight dip in the lift curve but continued to produce lift up to α = 50° for TI = 6.85% and 7.55%, respectively. This clearly shows that the turbulence intensity played a vital role in altering the surface flow characteristics thereby resulting in the change in the aerodynamic loads acting on the model. Therefore, it can be further added that the turbulence of the flow inflow influenced the stall characteristics of the trough incident LEP model as well. Based on the results, it can be reported that the turbulence intensity can effectively delay the stall for the 4° trough incident LEP model case considered in the present study for turbulence intensities ranging between 5.90% ≤ TI ≤ 7.55%. On the other hand, it should be noted that any further increase in the turbulence intensity could perturb the flow in such a way that it could negatively influence the surface flow characteristics of the trough incident LEP models thereby rendering detrimental effects on the lift characteristics curve. Beyond α = 50°, it can be seen from the figure that the time-averaged coefficient of lift kept on decreasing with the increasing of the angle of attack for all the TI cases.

Furthermore, to address the influence of the turbulence intensity on the variously modified trough incident LEP models, the 8° trough incident LEP model case was also considered and its corresponding time-averaged lift coefficient curves are displayed in Figure 5. As can be seen from the figure, with the increase in the trough incident angle, the influence of the turbulence intensity on the lift coefficient curve became adverse. For the 4° trough incident LEP model case, the time-averaged lift coefficient continued to increase from TI = 5.90% to TI = 7.55% even though similar results could be seen on the 8° trough incident LEP model where the lift coefficient was considerably less. For instance, it can be quantitively inferred from Figure 4 that the maximum lift coefficient for the 4° trough incident LEP model was 0.35, whereas for the 8° trough incident LEP model, the maximum lift coefficient was 0.33 which was 2% less. Also, it is evident from the 4° trough incident LEP model lift characteristics curve that it underwent complete stall between 40° ≤ α ≤ 50°, whereas for the 8° trough incident LEP model, the stall occurred at 20° ≤ α ≤ 30° for 5.90% ≤ TI ≤ 7.55%. From the results, it became clear that with the increase in the trough incidence angle, the effect of turbulence intensity also altered the stall characteristics. In other words, the 4° trough incident LEP model subjected to freestream turbulence intensities outperformed the 8° trough incident LEP model in terms of both the lift coefficient and the stall angle.

The effect of turbulence intensity on the drag characteristics of the 4° trough incident LEP model subjected to Re = 0.96 × 10^5^ is shown in Figure 6. It is worth noting that the drag values presented in this study were limited to pressure drag alone. It can be inferred from the figure that the drag coefficient (C_D_) increased with the increase in the turbulence intensity up to TI = 7.55% from 5.90%. Based on the observations of the experimental data, it was observed that the drag coefficient for the 4° trough LEP model subjected to lower magnitudes of turbulence was less influenced at smaller angles of attack i.e., in the pre-stall region, whereas, at higher angles of attack, the rate of increase of drag coefficient increased as a function of turbulence intensity. Based on the framework of the previous research, it has been identified that the average drag force increases as a function of turbulence intensity as outlined by Homann et al. [28]. As we know that the non-uniform flow characteristics prevalent over the LEP wings induce perturbations in the flow over the upper surface of the airfoil, when the turbulence intensity of the inflow increases, in addition to the perturbations induced by the LEP, the normalized perturbation of the mean velocity field adds to the existing perturbation which complicates the flow pattern resulting in a drag penalty. Simply, it can be reported that the perturbation induced by the turbulence intensity on the mean velocity field co-adjusts with the perturbation induced by the leading-edge protuberances and especially, the trough incidence angles incorporated on the test model. Furthermore, the drag coefficient of the 8° trough incident LEP model is also presented in this paper as shown in Figure 7. The figure shows that the drag coefficient decreased with the increase in the turbulence intensity throughout the regime for the 8° trough incident LEP model. The experimental results obtained for the 8° trough incident LEP model were in direct contrast with the 4° trough incident LEP model. In contrast to the 4° trough incident LEP model where the drag coefficient (C_D_) increased with the increase in the turbulence intensity up to TI = 7.55%, the 8° trough incident LEP model exhibited lesser drag for all the turbulence cases. From the figure, it is also evident that the 8° trough incident LEP model encountering freestream turbulence intensities of TI = 8.49% and 10.54% displayed the least drag coefficient when compared against TI varying between 5.90% ≤ TI ≤ 7.55%. To gain much deeper insight in to the reason behind the decrement in the drag coefficient for 8° trough incident LEP model, it becomes necessary to understand the underlying flow physics through surface pressure distribution which is subsequentially addressed in the later section for the higher Reynolds number.

(2)*Re* = *1.71* × *10*^5^

The coefficient of lift for the 4° trough incident LEP model operating at various turbulence intensities at a Re = 1.71 × 10^5^ is displayed in the Figure 8. It can be seen from the figure that for Re = 1.71 × 10^5^, the lift coefficient of the 4° trough incident LEP model subjected to a freestream turbulence intensity of TI = 5.90% increased linearly with the increase in the angle of attack up to α = 5°. This region is called the linear region beyond which it attains the maximum lift coefficient and then subsequently undergoes stall phenomenon. Compared with the TI = 5.90%, the other turbulence intensity cases like TI = 6.85, 7.55, 8.49 and 10.54% exhibit higher aerodynamic lift coefficient. For instance, at α = 5°, the 4° trough incident LEP model subjected to turbulence intensities of 6.85% ≤ TI ≤ 10.54% produced a lift coefficient of 0.08, 0.13, 0.09, and 0.09, respectively, whereas the same 4° trough incident of the LEP model subjected to a turbulence intensity of TI = 5.90% produced 0.05 which was slightly less than the lift coefficient of the remaining TI considered. At the same time, it should however be noted that all the TI cases except 5.90% and 10.54% exhibited a maximum lift coefficient at α = 25° whereas the remaining 4° trough incident LEP model subjected to a turbulence intensities 6.85% ≤ TI ≤ 10.54% exhibited stall at α = 25°. In addition to that, it can also be noted from the figure that the 4° trough incident LEP model subjected to a turbulence intensity of 6.85% ≤ TI ≤ 10.54% produced a maximum lift coefficient of 0.30, 0.30, 0.31, and 0.32, respectively, whereas the same 4° trough incident LEP model subjected to a turbulence intensity of TI = 5.90% and TI = 10.54% produced 0.32 which was slightly higher than the lift coefficient of the remaining TI considered. This difference in the lift performance characteristics of the same 4° trough incident LEP model implied that the aerodynamic loads acting over the trough incident models were also the function of the magnitude of the turbulence intensity like the conventional airfoil sections. In the pre-stall regime, even though the time-averaged mean coefficient of lift looked similar with some significant variation with respect to the increase in the TI, it is interesting to note that there were differences in the slope of the characteristic lift curves with the increase in the turbulence intensity and angles of attack. Likewise, difference in the stall delay angle were also observed with respect to the variation in the turbulence intensity.

Figure 9 illustrates the time-averaged coefficient of lift (C_L_) against the angle of attack (α) for the 8° trough incident LEP model at various turbulence intensities ranging from 5.90 to 10.5%. It is evident from the figure that the LEP model subjected to intensity TI = 7.55% produced a greater lift coefficient in the pre-stall region up to an angle of attack of α = 20° in comparison to all the other TI cases. Once it attained the maximum lift coefficient, the maximum lift coefficient gradually reduced with the increase in the angle of attack thereby resulting in a stall phenomenon. It can be seen from the figure that the 4° trough incident LEP model subjected to a TI of 5.90% experienced stall at α = 25°. However, the same 4° trough incident LEP model when subjected to a magnitude of higher turbulence rendered the aerodynamic benefit in terms of stall delay. For example, at TI = 6.85 and 7.55%, the same 4° trough incident LEP model underwent stall at α = 30° which was a 20% increase in the stall delay angle when compared to the lowest magnitude of turbulence case i.e., TI = 5.90%. It can be therefore inferred that the lift coefficient increased with the increase in the turbulence intensity near the vicinity of the stall region for the 4° trough incident LEP model up to TI = 7.55%. Aiming at identifying the influence of the Reynolds number, the lift coefficient curve of the 4° trough incident LEP model operating at Re = 1.71 × 10^5^ was compared against the Re = 0.96 × 10^5^. As indicated earlier, it can be observed from Figure 4 and Figure 8 that the lift coefficient increased with the increase in the Reynolds number. For instance, it can be observed from Figure 4 that at Re = 0.96 × 10^5^, the 4° trough incident LEP model subjected to a turbulence intensity of TI = 5.90% exhibited a lift coefficient of 0.28, whereas the same 4° trough incident LEP model subjected to the same turbulence intensity of TI = 5.90% exhibited 0.31 at Re = 1.71 × 10^5^ shown in Figure 8. At low angles of attack, i.e., in the pre-stall region, the 4° trough incident LEP model subjected to the turbulence intensities of 7.55% ≤ TI ≤ 10.54% produced slightly greater lift when compared to the TI = 5.90% and 6.85% cases up to α = 10°.

Moreover, towards the vicinity of the stall angle, the 8° trough incident LEP model subjected to turbulence intensities of 6.85% ≤ TI ≤ 10.54% produced more lift than the 4° trough incident LEP model subjected to a turbulence intensity of TI = 5.90%. This shows that the lift coefficient increased with the increase in the TI at higher angles of attack or near the vicinity of the stall region for the 8° trough incident LEP model. This holds in good agreement with the previous results as well indicating that the spanwise flow induced by the leading-edge protuberances at higher angles of attack was favorable while at pre-stall angles, especially up to α = 10°, it was disturbed by the perturbations induced by the turbulent flow. In other words, at smaller angles of attack, the perturbations induced by the turbulent flow could potentially affect the complex spanwise flow over the LEP model thus hindering the aerodynamic benefit. However, out of the two-step stall characteristics condition, at the second stall point, the LEP model subjected to the turbulence intensity of TI = 5.90% displayed a peak maximum lift coefficient in comparison to the other turbulence intensities. Wang et al. [29] reported that the decreasing slope at higher angles of attack towards the vicinity of the stall angle are linked to the separation followed by turbulent flow reattachment. It can be confirmed from the figure that the increase in the turbulence of the inflow kept the flow attached to the test models thereby resulting in delayed stall characteristics. For TI = 5.90%, the lift curve experienced a short dip at α = 30° whereas the same 8° trough incident LEP model subjected to a higher magnitude of turbulence continued to produce lift up to α = 50° with no significant dip in the lift curve.

The effect of turbulence intensity on the drag characteristics of the 4° trough incident model subjected to Re = 1.71 × 10^5^ is shown in Figure 10. It is evident from the figure that the rate of increase in the time-averaged mean drag coefficient increased remarkably at higher angles of attack when compared to the pre-stall angles. Figure 10 clearly shows that the 4° trough incident LEP model subjected to a turbulence intensity of TI = 5.90% experienced the highest amount of drag when compared to the other turbulence intensity cases considered for the present study. It is believed that with the increase in the turbulence intensity, the spanwise flow induced by the leading-edge protuberances gets disrupted by the oncoming freestream turbulent flow thus reducing the pressure drag acting over the same model by hindering the flow physics of LEP. To further ascertain this behavior and aiming at identifying the reason behind the change in the drag coefficient, the surface pressure distribution over the test model at TI = 5.90% and 6.85% were investigated as shown in Figure 11 for the same Re = 1.71 × 10^5^. From the figure, it can be clearly seen that that the flow over the peak section almost remained consistent with the increase in the TI. However, it is evident from the figure that with the increase in the TI, the flow over the trough section underwent significant change. In the case of TI = 5.90%, the negative suction pressure over the trough region clearly indicated that the majority of the flow went behind the trough region inducing an enhanced suction peak at the trough thereby drawing low-inertial boundary layer molecules from the peak, thus maintaining the non-uniform separation characteristics and thereby offering the aerodynamic benefit in terms of stall delay. However, in the case of TI = 6.85%, the increase in the turbulent flow hindered the spanwise flow induced by the LEP, resulting in the reduction in the negative suction pressure over the trough indicating that the non-uniform separation characteristics present over the general LEP cases gradually disappeared in this case. This clearly shows that the increase in the turbulent flow hindered the spanwise flow over the LEP airfoil thus complicating the flow pattern and eliminating the non-uniform separation characteristics along with the aerodynamic benefit. This plausibly explains the reason behind the decrease in the pressure drag to a certain level. The constant pressure region formed towards the vicinity of the trailing edge indicated the absence of the complete pressure recovery, but it is worth noting that the constant pressure region formed over the TI = 6.85% case was relatively less when compared to the TI = 5.90%. The effect of turbulence intensity on the drag characteristics of the 8° trough incident model subjected to Re = 1.71 × 10^5^ is shown in Figure 12. It can be seen from the figure that the rate at which the (c_d_) increased at higher angles of attack when compared to the pre-stall angles was slightly higher. It is evident from the figure that the 8° trough incident LEP model subjected to TI = 5.90% experienced the highest amount of drag compared to the other TI as discussed in the previous case of the 4° trough incident LEP model.

(3)*Re* = *2.48* × *10*^5^

Figure 13 shows the variation of the time-averaged coefficient of lift (C_L_) for the 8° trough incident model subjected to various turbulence intensities for different angles of attack ranging between 0° ≤ 90° at Re = 2.48 × 10^5^. The dashed lines represent the standard deviation of C_L_ while the solid lines represent the time-averaged C_L_. It is evident from the figure that the 8° trough incident model subjected to a freestream turbulence intensity of TI = 6.85% was lower than the TI = 5.90% case. It can be seen from the Figure 13a that at smaller angles of attack i.e., before the pre-stall angle, the 8° trough incident model subjected to lower magnitude of turbulence like TI = 5.90% produced a higher coefficient of lift when compared to the TI = 6.85% case. Consequently, from Figure 13b it can be noted that the higher magnitudes of turbulence TI = 7.55% produced a higher coefficient of lift compared to the baseline TI = 5.90% case. Likewise, it can be clearly seen from the figure that at α = 5°, there was a sudden drop in the coefficient of lift in the 8° trough incident model for the baseline test case, following which, the coefficient of lift regained positive value with the increase in the angle of attack. It is of interest to note that with the increase in the turbulence intensity, the negative lift produced by the trough incident model disappeared gradually. On the contrary, with the increase in the turbulence intensity beyond TI = 7.55%, it can be observed from the figure that the maximum lift coefficient decreased. Based on the framework of the previous studies, it is known that the LEP model exhibits smooth stall characteristics in opposition to the abrupt stall characteristics experienced by the conventional airfoil models. However, it is of interest to note that with the increase in the trough incidence angles, the LEP wing tended to behave like a conventional wing exhibiting abrupt stall characteristics. It is evident from the Figure 13c,d that for the 8° trough incident model subjected to TI ≥ 7.55% at α = 30°, an abrupt decrease in the lift coefficient can be seen. It can be seen from the Figure 13a that the 8° trough incident model subjected to TI = 5.90% stopped producing lift at α = 30° thus resulting in a stall phenomenon. The variation of standard deviation was utilized in this study as a measure of the dispersion of the pressure data. It has been identified from the experiments that in the post-stall region where the flow separation prevails, the dispersion of the data is widespread (indicated by the standard deviation) when compared against the pre-stall i.e., flow attached angles.

From the figure, it becomes clear that the trough incidence LEP model when exposed to the freestream flow featuring higher magnitudes of turbulence intensity, had exhibits modified stall characteristics. The dashed lines represent the standard deviation of C_D_ while the solid lines represent the time-averaged C_D_. For instance, it can be seen from Figure 13a that for the TI = 5.90% baseline case, the stall could be observed between α = 30°. On the other hand, the same test model when subjected to a freestream featuring turbulence intensity ranging between 6.85% ≤ TI ≤ 10.54% operating at Re = 2.48 × 10^5^, the phenomenon of the abrupt stall gradually disappeared. Furthermore, to ascertain that this phenomenon of abrupt stall behavior gradually decreased with the increase in the turbulence intensity, it can be confirmed from Figure 13a–d that for TI = 6.85%, the 8° trough incident model showed a drop in lift coefficient between 30° ≤ α ≤ 35°. For the same model, when subjected to higher magnitudes of turbulence TI ≥ 7.55% as shown in Figure 13c,d, it can be easily seen that the smooth stall behavior gradually disappeared and the abrupt loss of lift started occurring. Similar phenomenon of abrupt stall characteristics can be seen for the same 8° trough incident model subjected to a TI = 8.49 and 10.54% from Figure 13c,d. Therefore, it became clear that the abrupt stall characteristics observed in the trough incident LEP model decreased with the increase in the turbulence intensity, the onset of stall precedes this.

Figure 14a–d displays the variation of the time-averaged coefficient of drag (C_D_) and its standard deviation for the 8° trough incident model at Re = 2.48 × 10^5^. It is apparent from the figure that with the increase in the turbulence intensity, the coefficient of drag decreased especially at higher magnitudes of turbulence like TI = 8.49% and 10.54% considered in the present study. It can also be observed that the variation of the standard deviation in the pre-stall regime was relatively smaller than the post-stall region. As discussed in the lift performance characteristics, the noticeable change in the standard deviation at higher angles of attack was most likely due to the presence of the unstable flow induced alongside the separation. The constant pressure region observed from the surface pressure measurements clearly indicate that the pressure did not recover completely at the trailing edge. However, at higher magnitudes of turbulence like TI = 8.49% and 10.54%, the turbulence present in the oncoming flow energized the flow and thereby helped the flow attachment resulting in the lesser drag penalty. From Figure 14b at smaller angles of attack, the drag coefficient for the turbulent flow featuring TI = 7.75% was slightly higher than the TI = 5.90% case. It is speculated that this increase in drag is possibly caused by the additional perturbations caused by the turbulent nature of the inflow. However, at larger angles of attack, the momentum of the turbulent flow was utilized to keep the flow attached to the surface thus resulting in a lesser drag coefficient. It is observed from figure that maximum coefficient of drag subjected to TI = 5.90% is 0.82 and from TI = 7.55% was 0.79 at α = 90°.

Figure 15a,b clearly quantifies the force fluctuation amplitudes for the 4° and 8° trough incident model subjected to TI = 6.85%. It is clearly indicated from Figure 13a,b that the time-histories fluctuation of the coefficient of lift for α = 5° was relatively small when compared to the time-histories fluctuation of the (C_L_) measured at α = 25°. From this, it can be inferred that, at α = 5°, the magnitude of fluctuation of the (C_L_) was small as it falls in the linear region, free from the separated shear layers whereas at α = 25°, the magnitude of fluctuation was significantly larger in comparison to the α = 5° case. It can be construed that, as α = 25° was situated close to the separation point near the stall angle, the unstable flow induced by the separation might be the plausible reason behind the increase in the magnitude of the fluctuations. Similarly, Figure 15b represents the time histories of the coefficient of lift (C_L_) for the 8° trough incident model at α = 5° and α = 30° where it can be seen from the figure that the increase in the magnitude of the fluctuations observed with the increase in the angle of attack close to the separation point was valid in the 8° trough incident model case as well. Based on the comparison of the magnitude of fluctuations, it should however be noted that the amplitude of fluctuation for the 8° trough incident model was higher. Therefore, it can be reported that with the increase in the trough incident angle, the magnitude of fluctuations observed increased indicating that the unstable flow characteristics grew with the introduction of the incident angle.

## 4. Conclusions

Wind tunnel tests were performed to experimentally evaluate the influence of turbulence intensity on the trough incident LEP models at a wide range of angles of attack ranging from 0° to 90° with an increment of 5° for four different Reynolds number in the range of 10^5^. Based on the experimental investigation, the following conclusions were made:The aerodynamic forces acting over the LEP model changed with the change in the trough incidence angle. This clearly showed that the trough incidence angle had a significant influence on the aerodynamic characteristics of the LEP model and needs to be included as a primary factor next to amplitude and wavelength.The 8° trough incident model exhibited the maximum coefficient of lift when compared to the 4° trough incident LEP model, signifying that with the increase in the trough incidence angle, the maximum coefficient of lift increased.It can be inferred that the turbulence intensity had a significant effect on the aerodynamic characteristics of trough incident LEP models. It is evident that with the increase in the turbulence intensity, the drag value reduced for both the 4° and 8° trough incident LEP models particularly at post-stall angles.Results indicated that increasing turbulence intensity at pre-stall angles of attack led to an increase in the lift coefficient for trough incident LEP models. However, beyond post-stall angles, higher turbulence intensity became detrimental due to the complex flow induced by the trough incidence angles. Notably, at the 8° trough incidence angles, an increase in turbulence intensity resulted in a reduction in the lift coefficient at post-stall angles. This highlighted the complex interplay between turbulence intensity, trough incidence angle, and the angle of attack on aerodynamic performance.Surface pressure distribution prevailing over the trough incident LEP still exhibited the non-uniform separation characteristics amidst the complications induced by the turbulence and the incidence angle.

## Figures and Tables

**Figure 1 biomimetics-09-00354-f001:**
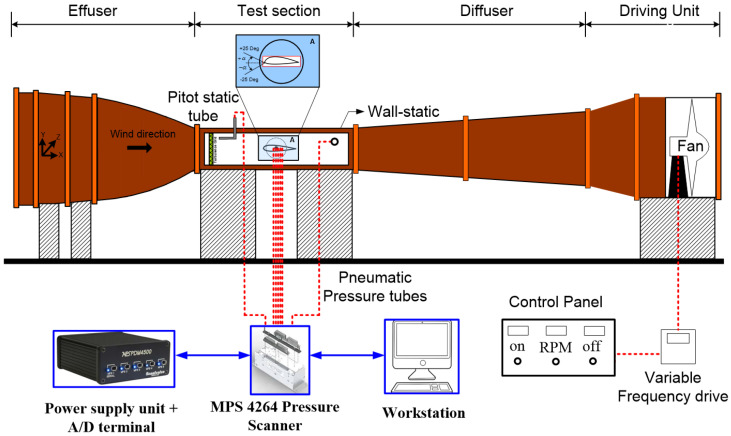
Experimental setup, wind tunnel.

**Figure 2 biomimetics-09-00354-f002:**
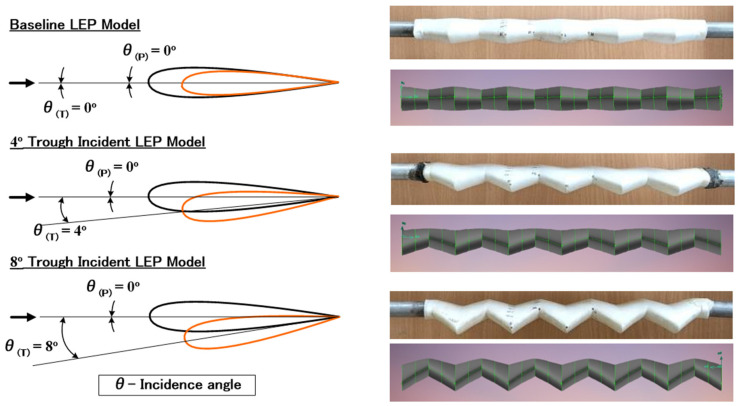
Schematic representation of the baseline LEP and the trough incident LEP Models.

**Figure 3 biomimetics-09-00354-f003:**
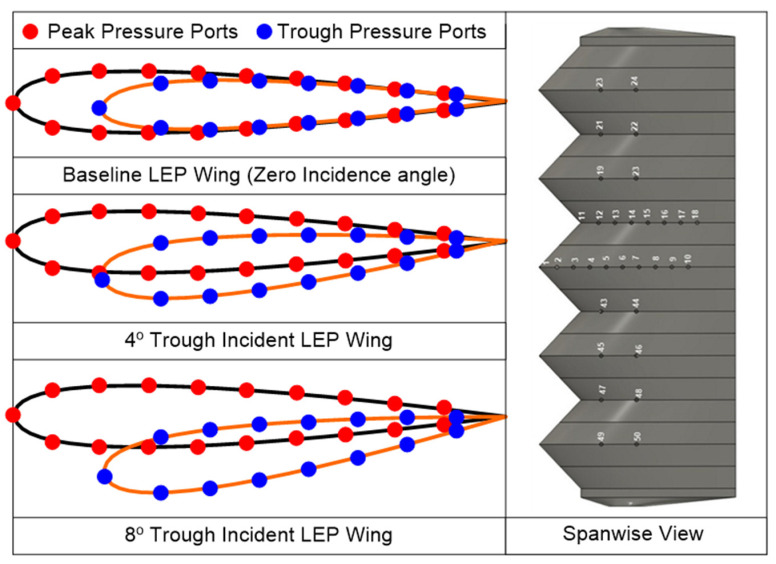
Schematic representation of the location of pressure taps on trough incident LEP model.

**Figure 4 biomimetics-09-00354-f004:**
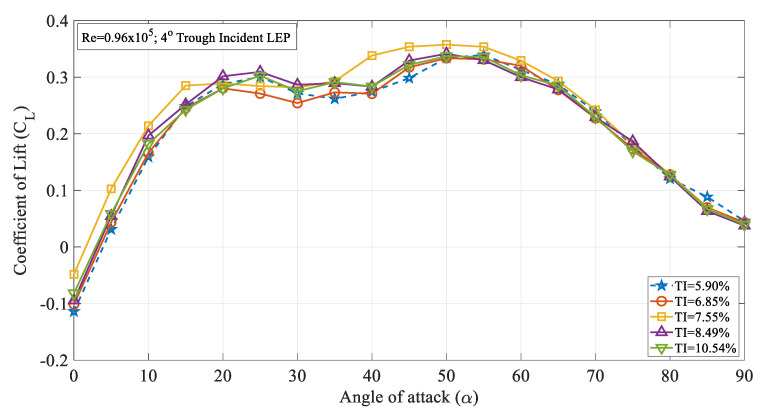
Time-averaged coefficient of lift (C_L_) vs. angle of attack (α) for the 4° trough incident model at Re = 0.96 × 10^5^.

**Figure 5 biomimetics-09-00354-f005:**
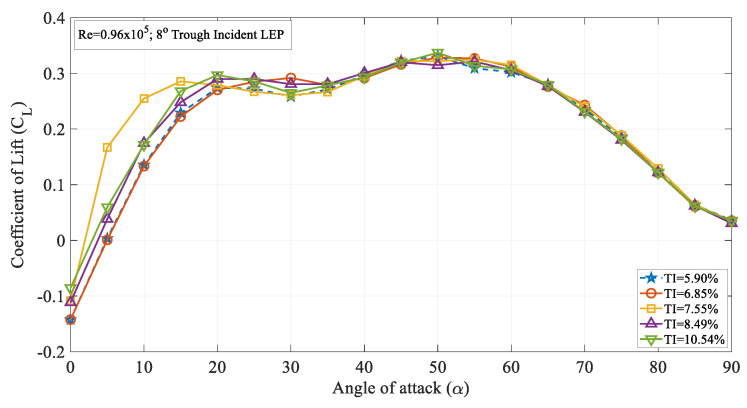
Time-averaged coefficient of lift (C_L_) vs. angle of attack (α) for the 8° trough incident model at Re = 0.96 × 10^5^.

**Figure 6 biomimetics-09-00354-f006:**
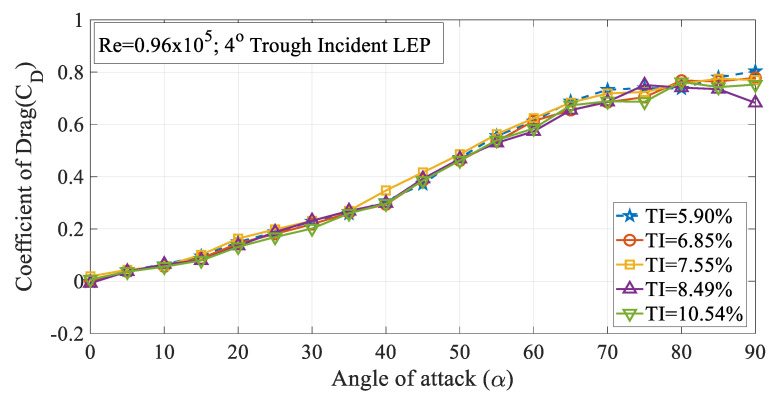
Time-averaged coefficient of drag (C_D_) vs. angle of attack (α) for the 4° trough incident model at Re = 0.96 × 10^5^.

**Figure 7 biomimetics-09-00354-f007:**
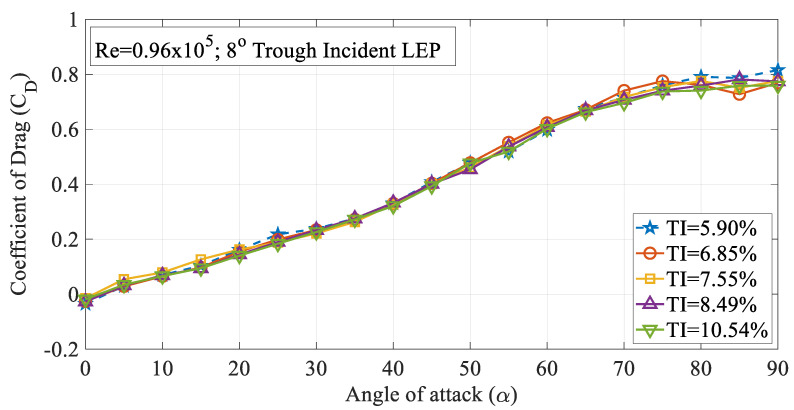
Time-averaged coefficient of drag (C_D_) vs. angle of attack (α) for the 8° trough incident model at Re = 0.96 × 10^5^.

**Figure 8 biomimetics-09-00354-f008:**
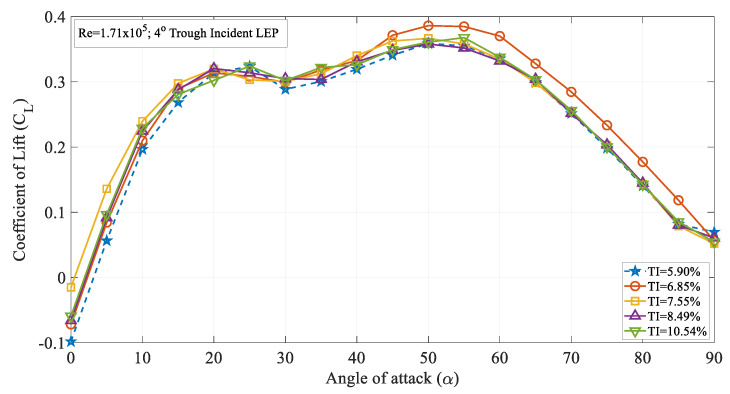
Time-averaged coefficient of lift (CL) vs. angle of attack (α) for the 4° trough incident model at Re = 1.71 × 10^5^.

**Figure 9 biomimetics-09-00354-f009:**
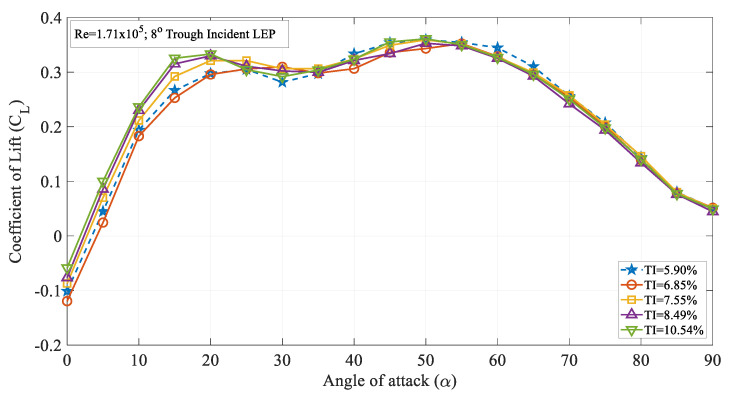
Time-averaged coefficient of lift (C_L_) vs. angle of attack (α) for the 8° trough incident model at Re = 1.71 × 10^5^.

**Figure 10 biomimetics-09-00354-f010:**
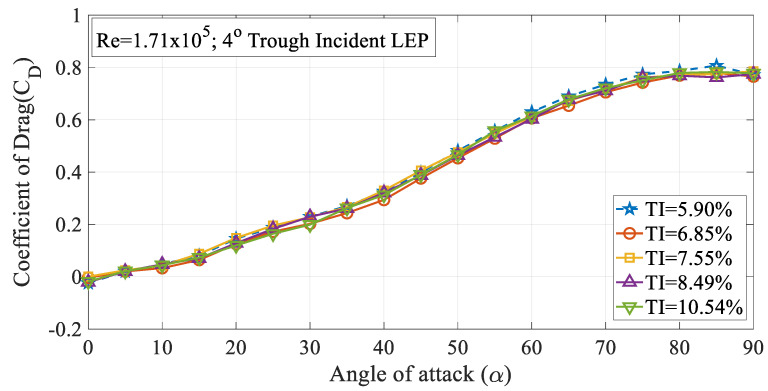
Time-averaged coefficient of drag (C_D_) vs. angle of attack (α) for the 4° trough incident model at Re = 1.71 × 10^5^.

**Figure 11 biomimetics-09-00354-f011:**
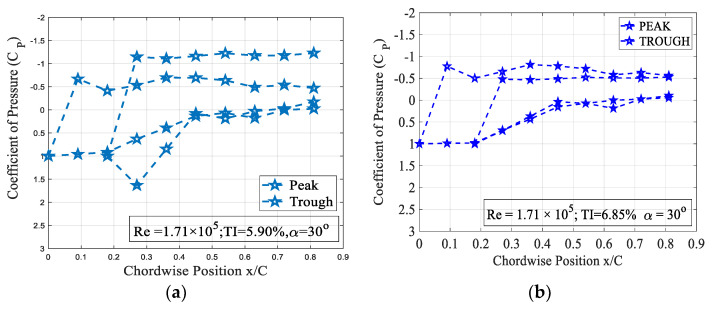
Surface pressure distribution of the 4° trough incident LEP model subjected for Re = 1.71 × 10^5^ at α = 30° (**a**) TI = 5.90% (**b**) TI = 6.85%.

**Figure 12 biomimetics-09-00354-f012:**
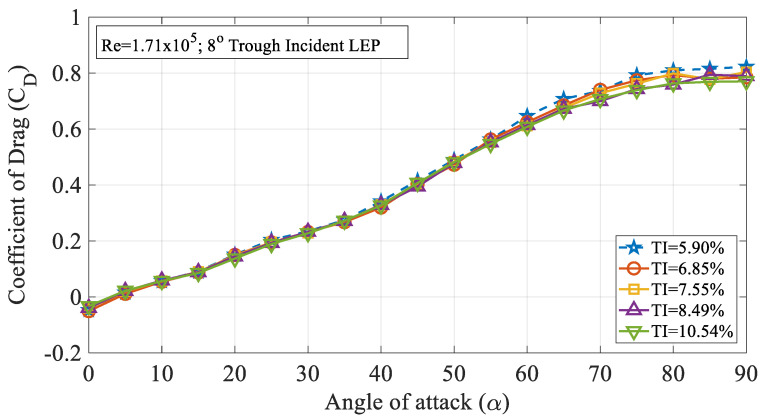
Time-averaged coefficient of drag (C_D_) vs. angle of attack (α) for the 8° trough incident model at Re = 1.71 × 10^5^.

**Figure 13 biomimetics-09-00354-f013:**
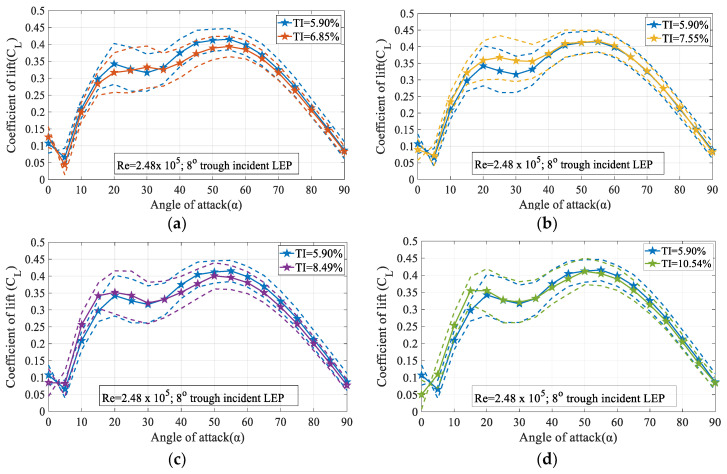
Time-averaged coefficient of lift (C_L_) vs. angle of attack (α) for the 8° trough incidence model at Re = 2.48 × 10^5^ (**a**) for TI = 5.90% & 6.85% (**b**) for TI = 5.90% & 7.55% (**c**) for TI = 5.90% & 8.49% and (**d**) for TI = 5.90% & 10.54%.

**Figure 14 biomimetics-09-00354-f014:**
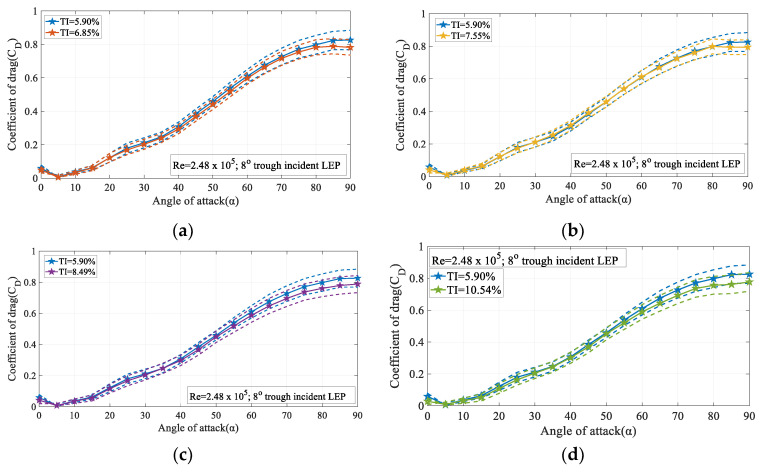
Time-averaged coefficient of drag (C_D_) vs. angle of attack(α) for the 8° trough incidence model at Re = 2.48 × 10^5^ (**a**) for TI = 5.90% & 6.85% (**b**) for TI = 5.90% & 7.55% (**c**) for TI = 5.90% & 8.49% and (**d**) for TI = 5.90% & 10.54%.

**Figure 15 biomimetics-09-00354-f015:**
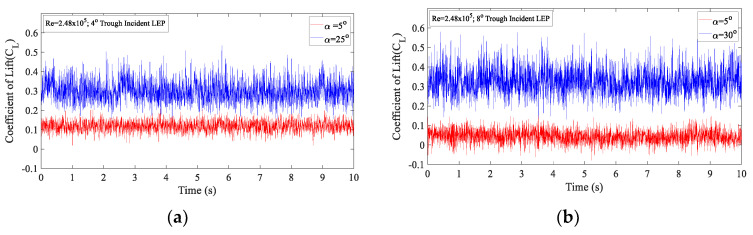
Time histories of the lift coefficient (C_L_) for TI = 6.85% at Re = 2.48 × 10^5^ (**a**) 4° trough incident model at α = 5° and α = 25° (**b**) 8° trough incident model at α = 5° and α = 30°.

**Table 1 biomimetics-09-00354-t001:** Geometrical Parameters of the test model.

Label	A/c	λ/c	$\theta_{P}$	$\theta_{T}$
Baseline	0.12	0.25	0	0
Trough_Inc_4deg	0.12	0.25	0	4
Trough_Inc_8deg	0.12	0.25	0	8

## Data Availability

Dataset available on request from the authors.

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
