# Peer review of "Effect of Trough Incidence Angle on the Aerodynamic Characteristics of a Biomimetic Leading-Edge Protuberanced (LEP) Wing at Various Turbulence Intensities"

_biomimetics, 2024, doi:10.3390/biomimetics9060354_

Round 1
Reviewer 1 Report (Previous Reviewer 1)
Comments and Suggestions for Authors
I reviewed this manuscript about a year back. I looked at my comments then, and the present manuscript has explained all the queries raised.
I find the work is presented well with the required details. I would recommend the manuscript to be accepted as it is.
Author Response
The author(s) express their sincerest thanks to the reviewer for providing constructive comments. We would like to thank you for your time and efforts in augmenting the quality of this manuscript.

Reviewer 2 Report (Previous Reviewer 3)
Comments and Suggestions for Authors
I accept the paper in the improved version.
Author Response
The author(s) express their sincerest thanks to the reviewer for providing constructive comments. We would like to thank you for your time and efforts in augmenting the quality of this manuscript.

Reviewer 3 Report (New Reviewer)
Comments and Suggestions for Authors
The authors study the aerodynamic performance of a wing with Leading Edge
Protuberances (LEP). They analyze the lift and drag forces from pressure
measurements for a range of angles of atack (0-90 degrees) subjected to
different levels Turbulence Intensity (TI) present in the free stream.
The paper is well written, however it could benefit from editing regarding
the English/presentation. The measurements seem trustworthy and accurate, but
the main objective of the work seems not clear and the convey of results lacks
of clarity. Therefore, I do not recommend the manuscript for publication in
the journal Biomimetics because of the concerns I detail below.
My major concern is that the main message of the manuscript is not clear.
There is a lot of analysis on the turbulence intensity, plus the authors
claim that there is an influence of TI on the results. However, the results
seem pretty insensitive to TI (the differences being inside the variation of
the signal, see figures 13 and 14), and indeed there is no mention to the effect
of TI in the conclusions. Furtheremore, the base case with TI=0 is missing
in order to infer any conclusion of the observations with TI>0.
Another major concern is the analysis of three different Re, not so different
from each other, to infer more insight about the problem. Do the authors
vary the Re number to analyze the effect on the TI influence, or the
different geometries?
Regarding the interpretation of the results, the authors use the term
linear in an arbitrary fashion. They first say that the CL grows linearly
until AoA 20,25 (lines 416, 424) and then until AoA 5 for a similar
behavior (line 495).
Finally, I add the following minor comments:
- Language seems sometimes colloquial ("till" is used in several occasions).
- Abstract: attempts? there is no attempt, simply a description of the results
- Line 48: Do the authors mean "ocean conditions" instead of "atmospheric conditions"
- Line 59: by10% is together.
- Line 71 : acts -> act
- Line 80: At `10**5` ...., missing what quantity is `10**5` )
- Line 99: Confusing quantities (0.12c or 0.12%c)?
- Line 441: Missing dot to split sentences.
- Line 606: Figure 13 containes dashed lines that are not described, nor mentione anywhere.
- Symbol theta and alpha is used for AoA.
Comments on the Quality of English Languageminor changes like acts -> act. Otherwise seems correct.
Author Response
The author(s) express their sincerest thanks to the reviewer for providing constructive comments. We would like to thank you for your time and efforts in augmenting the quality of this manuscript.
The author(s) have addressed all the valuable comments provided by the reviewer individually and hereby submit the review response for your kind reference.
Kindly advise us.

This manuscript is a resubmission of an earlier submission. The following is a list of the peer review reports and author responses from that submission.
Round 1
Reviewer 1 Report
Comments and Suggestions for Authors
1. It is understood that the authors are clear with the literature and nature of geometry for LEP airfoil configuration. But for the reader, it would be good to present the geometry of the base LEP wing and modified wings in a figure.
2. Figure 2, the title should be 4 deg Trough
3. A general query: why did the authors not compare results of 4 deg and 8 deg with the base case?
4. How the authors achieve TI variation has not been detailed. Please do that.
5. It will be good to represent the location of pressure taps in a figure (either of the suction side or pressure side )
6. fig 10 and 11 naming is wrong
7. why do authors use the dotted line for Figure 10 onwards (is this because fluctuations are high with higher Re)
8. The comparison of Cl characteristics across the Re range should be done to get a comprehensive presentation of the work.
Comments on the Quality of English Language
1. The manuscript has to be presented in the template of the journal.
2. In Certain places the figure titles are wrong. Kindly correct them.
Reviewer 2 Report
Comments and Suggestions for Authors
This paper tests the aerodynamic performance of a wing with Leading-Edge Protuberanced in different turbulence intensities through wind tunnel experiments. The results can be a reference for other researchers. However, the paper needs to be revised substantially before it can be accepted:
1. The first paragraph of the Introduction is too long for the reader. The authors are requested to split the first paragraph. In addition, there is insufficient correlation among the paragraphs of the introduction part, especially the penultimate paragraph of the introduction is inappropriate in this location. This is because the author introduces the effect of turbulence intensity before this paragraph, but this paragraph seems to have little relevance to the topic of the paper. The author needs to summarize the story of the introduction better.
2. Table 1 is not mentioned in the main text.
3. The authors should at least provide CAD models or pictures of the experiments used for this article, otherwise it is not even clear to the reader what the experiments are about.
4. The authors are requested to add pictures or schematics of the experimental models with pressure taps installed so that the reader can easily reproduce the experiment.
5. The captions of Fig. 10 and Fig. 11 are incorrectly placed. Also, the title of the vertical coordinate of Fig. 10 seems to be wrong, it should be drag coefficient?
Reviewer 3 Report
Comments and Suggestions for Authors
The paper deals with the behaviour of the lift and drag coefficients of a NACA 63(4)-21 wing with LEP at different angles of attack, inflow turbulence and Re numbers.
It gives an overview of the pressure measurements describes the measured results. For a publication in a technical journal, the scientific content of the investigations is very limited. From the reviewer's point of view, it has the form of an experimental report.
In the introduction, reference is made to the biological background and a large number of investigations are mentioned that deal with the implementation of the findings on an aerofoil. These explanations lack a chronological structure. At the end it remains unclear what the authors want to contribute in their investigations in terms of new findings or which unsolved questions they want to solve.
The investigations are carried out in a small wind tunnel. Here, the first important thing would be to compare the measured pressure distribution without LEP with the values in the literature for the NACA 63 profile. It gives an insight into the measurement accuracy of the experimental setup as a first step. It also completely lacks an error analysis.
The exact description of the LEP also remains unclear. Here a sketch with all parameters is important. There is little information about the inflow conditions. TI is only one parameter. How and where was it measured? With which measuring technique was the investigations carried out. Is it an averaged parameter over the entire cross-section? What are the other turbulence parameters? The anisotropy of the turbulence and the length scales also have a very large influence on the experimental results.
Important parameters are also missing in relation to the pressure measurements. For the determination of the frequency resolution, the length of the pressure pipes is essential. What are the dimensions of the pressure holes. Again, the size (diameter) of the pressure holes has a critical influence on the measurement results.
These are some fundamental questions that must be clarified at the beginning. Only then is a scientific interpretation of the measurement results possible.
In addition, the significance for the physical understanding based only on pressure measurements is very limited. It would be important to have additional information about the velocity field in order to be able to achieve a verifiable interpretation of the investigations.
In addition, there are other effects of the laminar-turbulent boundary layer transition that overlap with the effect of the LEP, so that it is difficult to recognise a general validity of the investigations from the results.
There are still a number of minor typing errors in the publication. The layout of the diagrams generally needs to be improved. Some important literature on this topic is also missing.
So that, unfortunately, my decision leads to a rejection of the publication. Fundamental improvements are needed, which require a continuation of the experimental investigations and result in a complete revision of the publication.
Reviewer 4 Report
Comments and Suggestions for Authors
There are remarks and marks in the document.
A picture of the setup would be great.
The models are printed in plastic and there is not a single image of the models, to make the reader easily visualize the trough angle.
Instrumentation of the wing models with 50 pressure taps is also of great interest in the experimental aerodynamics community and therefore it can be shown. Thus, pattern of taps may be visible. Integration of tubing is again of interest.
Trough incidence angle: there is no scheme to see the 4 and 8 deg models.
There is no force measurement, unfortunately. That may produce more certain results.
Sometimes there are unphysical results:
Cpmax>1
CL at AoA=0 deg present spikes - Fig 11
Reynolds numbers are quite small when compared to the industrial relevant wind turbines.
The self-citation can be removed.

Some improvements can be done, please see the attachment.